# Video Causal Understanding with Scene-conditioned Counterfactuals

## Abstract

Understanding the causal consequences of actions is critical for developing reliable embodied agents. A key challenge is attributing an observed outcome to a particular action within a video. Existing video analysis methods often rely on correlation and struggle to distinguish true causation from spurious association, as they do not explicitly model confounding factors. To address this, we reframe the task as a retrospective counterfactual inquiry, which allows us to quantify an action's necessity for an outcome. We then propose an efficient and doubly robust estimator that adjusts for confounding variables learned from video frames, providing resilience against model misspecification. To validate our approach, we conduct experiments in a controlled environment. The results show that our method provides more accurate causal attribution compared to baselines.

## 1 Introduction

The goal of creating general-purpose agents that can operate in varied human environments requires an ability to adapt to new situations Gupta et al. (2021); O'Neill et al. (2024). Recent theory suggests that for an agent to be robust, it must learn an underlying causal model of its environment rather than just surface-level statistics Richens & Everitt (2024); Gupta et al. (2024). This need is especially acute in settings like homes, where each space has unique layouts and routines. To generalize effectively, an agent cannot rely on a universal causal model; it must be able to infer the specific causal rules governing a particular scene.

To illustrate, consider an agent observing a person press a button near a front door, after which the door opens. The agent might infer a direct causal link: *button → door opens*. However, the true causal structure could be different: the button only controls the lights (*button → lights on*), while the door is motion-activated (*person's movement → door opens*). This spurious association is revealed only by exceptions, such as when someone exits without pressing the button, yet the door still opens. Distinguishing habit from cause in this single scene is crucial for generalization. Our goal is to enable an agent to answer a scene-conditioned counterfactual question: in that particular scene, had a key action been different, how would the outcome have changed?

Many studies have explored reasoning from video. Early work on perceptual causality used information-theoretic metrics to identify simple causal events Fire & Zhu (2016). More recently, data-driven methods have used end-to-end models to discover associations between events in video Chen et al. (2024b); Liang et al. (2022). Another line of research focuses on learning latent causal representations from video data Chen et al. (2024a); Wang et al. (2024). However, these approaches often do not explicitly account for confounding. Modeling confounding in video is difficult because of a fundamental non-identifiability problem. The visual scene before an action often influences both the action taken and the final outcome (*scene → action*, *scene → outcome*). This creates a spurious path that can be mistaken for a direct causal link, making it difficult to isolate the true effect of the action from observational data alone without additional assumptions Pearl (2009); Hernán & Robins (2020).

In this paper, we propose a framework to address this challenge by inferring scene-conditioned counterfactuals from observational video. Our key idea is that the temporal structure of video provides the necessary assumption to overcome the non-identifiability problem. By conditioning on the rich visual state contained in the frames *before* an action, we can block the confounding path between the scene and the outcome, allowing for a valid causal estimate. First, we provide a formal

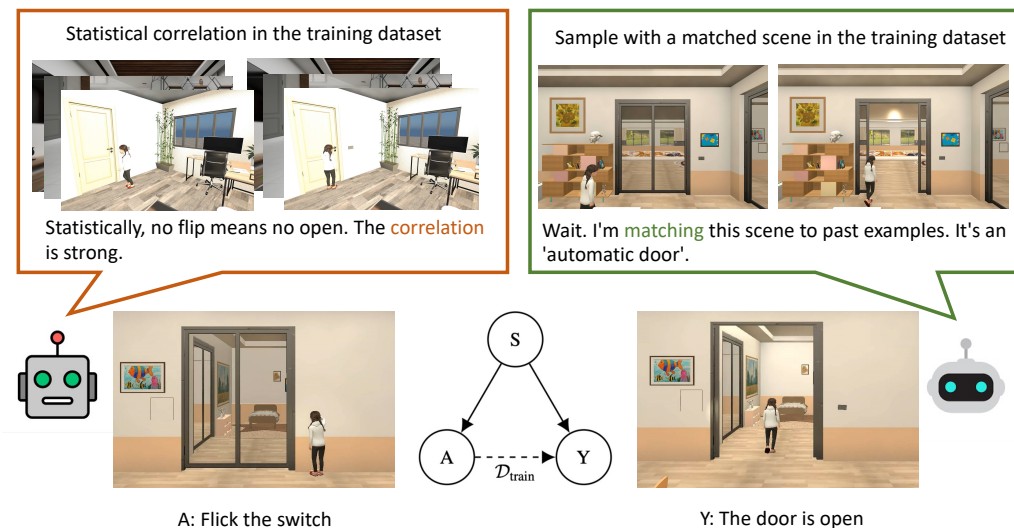

Figure 1: **Scene-Specific Causation: Beyond Training Set Correlation.** In the depicted scene, the **Action** ($A$) of pressing the light switch does not cause the **Outcome** ($Y$) of the door opening. The **Scene** ($S$)—which includes the person's movement and the door's motion sensor—is the true cause of the door opening. However, a correlation model trained on $\mathcal{D}_{\text{train}}$ might learn a spurious link ($A \xrightarrow{\mathcal{D}_{\text{train}}} Y$), because in some training scenes, pressing a button does open a door, leading to a generalization error. Our work infers the scene-specific causal graph $\{S \rightarrow A, S \rightarrow Y\}$ for this particular instance, correctly identifying that $A \nrightarrow Y$.

problem formulation for estimating event-level, scene-conditioned counterfactuals. Second, we develop an efficient estimation procedure built on two components: an outcome model that predicts the result under a counterfactual action, and a propensity model that estimates the probability of the observed action. To improve the reliability of our estimates, the procedure is doubly robust, meaning it remains consistent if either the outcome or the propensity model is correctly specified. Finally, to extract the semantic scene variables that the estimator operates on, we use modern foundation models to interpret the pre-action frames.

Our contributions are as follows:

1. We formulate the problem of inferring scene-conditioned counterfactuals from video.
2. We propose an efficient and doubly robust estimator to solve this problem.
3. We validate our approach in a controlled simulation with extensive experiments.

## 2 RELATED WORKS

**Foundations: identification targets and estimators.** Classical work in the potential outcomes framework Neyman (1923); Rubin (1974) splits along two identification routes for observational data. Cross-sectional adjustment controls observed confounders by outcome modeling or propensity-score balancing Rosenbaum & Rubin (1983), typically targeting average effects or individual predictions given rich covariates. Longitudinal adjustment composes conditional outcome models along histories via the g-formula Robins (1986). Both rely on plug-in estimation and are sensitive to outcome-model misspecification, especially with high-dimensional features. Semiparametric estimators address this by combining outcome and selection models to obtain doubly robust consistency and influence-function–based efficiency Robins et al. (1994); Scharfstein et al. (1999); Funk et al. (2011); Tsiatis (2006); van der Laan & Rose (2011); Bickel et al. (1993). Our target differs in scope but not in logic: we ask for a *scene-conditioned, event-level* counterfactual within a fixed episode and adopt the same backdoor principle (adjust on the pre-action scene), but avoid pure plug-in estimation by using influence-function corrections for stability

**Counterfactual reasoning: matching, representation, and retrospective queries.** Three strands organize counterfactual prediction under selection on observables. Matching-based methods compare treated and control units with similar covariates; they reduce modeling reliance but suffer when overlap is poor or dimension is high Rosenbaum & Rubin (1983). Representation-learning methods learn balanced embeddings across actions to support individual counterfactual prediction; they improve fit but can conceal residual imbalance or induce sensitivity to representation choices Johansson et al. (2016); Shalit et al. (2017); Louizos et al. (2017); Zeng et al. (2020). A third strand studies retrospective queries that condition on the observed outcome (causes-of-effects style). These quantify whether an action caused the observed effect rather than predicting the effect under an alternate action, and often require stronger structural assumptions than forward-looking queries. In contrast, our query is prospective in form but *episode-specific*: we condition on the factual scene, model action endogeneity, and combine prediction with a selection model to control bias while retaining individual-level resolution.

**Video causal reasoning: task families and assumptions.** Prior video work falls into four families with distinct problem settings. Perceptual causality studies local physical triggers (e.g., collisions) over short horizons and assumes compact mechanisms visible in a few frames Fire & Zhu (2015). Causal video QA benchmarks (e.g., CLEVRER, CoPhy) probe causal and counterfactual questions in controlled worlds; methods rely on object-centric state and scripted reasoning but face limited open-world variability Yi et al. (2020); Baradel et al. (2020). Multi-event discovery seeks event-level graphs or latent causal processes over long sequences, emphasizing structure recovery rather than calibrated counterfactuals for one episode Chen et al. (2024b;a). Embodied-evaluation regimes argue that robust behavior requires understanding world regularities and, under distribution shift, causal structure Peng et al. (2024); Richens & Everitt (2024). Compared to these, our problem is narrower but more quantitative: after a specific event in a real scene, estimate the outcome under an alternate action. This demands scene-conditioned adjustment and estimators that remain valid under imperfect visual models, rather than only logic chaining or graph recovery.

**Scene representations: structured extraction, discriminative features, generative latents.** Pre-action scene state can be modeled in three ways, each with trade-offs for adjustment. Structured extraction turns pixels into entities and attributes via promptable segmentation and open-vocabulary detection; it yields transparent conditioning variables but can miss fine-grained or unlabeled factors Kirillov et al. (2023a); Ravi et al. (2024); Liu et al. (2024a). Discriminative self-supervised features provide strong semantics and transfer but lack calibrated likelihoods, limiting principled uncertainty handling. Generative latents (e.g., VAEs and temporal variants) offer probabilistic structure and compression but may misalign with semantic factors without additional supervision Kingma & Welling (2013); Yao et al. (2022); Song et al. (2023). Causal representation learning argues for aligning latents with generative factors to improve robustness Schölkopf et al. (2021), while multi-modal models can supply high-level scene summaries but may inherit biases Comanici et al. (2025); Wang et al. (2025). Our estimator uses a hybrid state—structured variables for known entities and compact latents for residuals—so that conditioning remains interpretable while models stay expressive for efficient influence-function estimation (§4).

## 3 PROBLEM FORMULATION

In this section, we provide a formal mathematical definition for the retrospective counterfactual inference task introduced in the introduction, and specify the target quantities our method estimates.

**Problem setup.** Consider a short household interaction video $V$, with an outcome $Y \in \mathcal{Y}$ measured at a fixed time horizon after an event. Let $V_0$ be a fixed window of pre-action frames describing the scene. From $V$, we extract a sequence of discrete actions $A = (A_1, \ldots, A_K)$ that occur before the outcome. We are interested in the causal relationship between a specific target action $A^i$ and the outcome $Y$ within this particular episode. We denote the remaining actions by $A^{-i} = (A_1, \ldots, A_{i-1}, A_{i+1}, \ldots, A_K)$. For any action $a \in \mathcal{A}_i$, let $Y^{(i)}(a)$ be the potential outcome that would have been observed had the target action been set to $A^i = a$, holding the realized episode context $(V, A^{-i})$ constant. The observed outcome is therefore $Y = Y^{(i)}(A^i)$. Our objective is to infer, based on the observed scene and co-actions of this episode, what the outcome would have been had the target action been different.

**Event-level objective.** Our analysis is conditioned entirely on a single, realized event. As motivated in the introduction, our central goal is to answer a "what if" question for a specific scene. We formalize this as the scene- and co-action-conditioned counterfactual mean:

$$\psi_i\big(a' \mid y_0, a_0, a^{-i}, v\big) = \mathbb{E}\Big[Y^{(i)}(a') \,\Big|\, Y = y_0,\, A^i = a_0,\, A^{-i} = a^{-i},\, V = v\Big], \qquad (1)$$

where $(y_0, a_0, a^{-i}, v)$ is the fully realized episode and $a'$ is an alternative value for the target action. Equation equation 1 precisely poses the question: given everything that occurred in this specific episode, what would the expected outcome be if the target action were $a'$ instead of $a_0$? For binary outcomes, this allows us to compute the scene- and co-action-conditioned switch contrast:

$$\tau_i\big(a_0 \to a' \mid y_0, a^{-i}, v\big) = \mathbb{E}\Big[Y^{(i)}(a_0) - Y^{(i)}(a') \,\Big|\, Y = y_0,\, A^i = a_0,\, A^{-i} = a^{-i},\, V = v\Big], \quad (2)$$

which captures the expected change in the outcome from hypothetically switching the action from $a_0$ to $a'$ within the identical context.

Unlike average or conditional average treatment effects (ATE/CATE) that marginalize over covariate distributions Neyman (1923); Rubin (1974); Shalit et al. (2017), our estimand is retrospective and scene-conditioned on a realized episode: it asks for the counterfactual outcome in *this* scene given the realized co-actions. This distinction aligns with Pearl's counterfactual layer on the causal ladder and requires explicit structural assumptions for identifiability Pearl (2009).

**Assumptions.** Our analysis is built upon a set of causal assumptions. These assumptions formalize the structural properties of video data discussed in the introduction, providing the theoretical foundation for the methodology that follows.

**Assumption 3.1** (Stable unit treatment value). *There is no interference between episodes. The potential outcome $Y^{(i)}(a)$ is well-defined for each action $a$, and the observed outcome is $Y = Y^{(i)}(A^i)$.*

**Assumption 3.2** (Positivity). *For any context $(v, a^{-i})$ that occurs with non-zero probability, and for each action $a$ of interest, there is a non-zero probability of that action being taken: $P\big(A^i = a \mid V = v, A^{-i} = a^{-i}\big) > 0$.*

**Assumption 3.3** (Ignorability with scene and co-actions). *There exists an exogenous scene state $S$ at the time of the event such that, given $S$ and the co-actions $A^{-i}$, the choice of the target action $A^i$ is independent of the set of all potential outcomes:*

$$\big\{ Y^{(i)}(a) : a \in \mathcal{A}_i \big\} \;\perp\!\!\!\perp\; A^i \mid (S, A^{-i}).$$

**Assumption 3.4** (Potential outcomes depend only on scene and co-actions). *The potential outcomes are conditionally independent of each other given the scene state and co-actions: for all $a, a' \in \mathcal{A}_i$,*

$$Y^{(i)}(a) \;\perp\!\!\!\perp\; Y^{(i)}(a') \mid (S, A^{-i}).$$

**Assumption 3.5** (Scene–video separability). *The video $V$ serves as a measurement of the scene state $S$. It provides no additional information about the outcome or actions beyond what is contained in $S$:*

$$V \;\perp\!\!\!\perp\; (Y, A^i, A^{-i}) \mid S.$$

Assumptions theorems 3.1 and 3.2 are standard regularity conditions. Practically, the positivity requirement may be strained by rare actions or safety-critical behaviors in video logs; diagnostics and design choices to mitigate weak overlap are well-studied in causal epidemiology and apply here as well Petersen et al. (2012). The key causal assumptions are theorems 3.3 and 3.4, which together state that the latent scene state $S$ and observed co-actions $A^{-i}$ are sufficient to account for all confounding between the target action and its outcomes. Finally, theorem 3.5 provides the crucial link between theory and data, formalizing the idea that the pre-action video $V_0$ is our observational window into the underlying scene state $S$.

## 4 METHODOLOGY

In this section, we develop the estimation strategy for the target quantity defined in Equation equation 1. In Section 4.1, we first address the challenge of estimating this counterfactual quantity from

observational data. We then derive an identifiable expression based on the assumptions from the previous section and use it to construct a statistically efficient estimator, discussing its connections to and differences from existing literature. In Section 4.2, we detail how to learn the necessary components of this estimator from video data. For clarity, symbols with hats denote learned or estimated quantities (e.g., $\hat{\psi}_i$), while those without denote their population counterparts (e.g., $\psi_i$).

## 4.1 ESTIMATING SCENE-CONDITIONED COUNTERFACTUALS

Our goal is to estimate $\psi_i\big(a' \mid y_0, a_0, a^{-i}, v\big)$. A core challenge is that we only observe factual events; counterfactual outcomes are never directly observed. Consequently, we cannot simply train a supervised model to predict $\psi_i$, as the ground-truth labels are missing. In general, without additional structural assumptions, this quantity is not identifiable from observational data.

However, the assumptions we laid out in Section 3 provide the necessary structure for identification. They allow us to express our target quantity as an integral over the distribution of observable data. First, we define the retrospective event of interest:

$$\mathcal{E}_{\text{fact}} := \{Y = y_0,\ A^i = a_0,\ A^{-i} = a^{-i},\ V = v\}, \qquad e_{\text{fact}} := P(\mathcal{E}_{\text{fact}}) > 0.$$

Based on our assumptions, the target counterfactual quantity can be identified as:

$$\psi_i\big(a' \mid y_0, a_0, a^{-i}, v\big) = \int \underbrace{\mathbb{E}\big[Y \mid A^i = a',\ A^{-i} = a^{-i},\ S = s\big]}_{\mu_i(a' \mid a^{-i}, s)}\, p(s \mid \mathcal{E}_{\text{fact}})\, ds. \tag{3}$$

This expression embodies an intuitive matching idea. It first infers the posterior distribution of plausible scene states, $p(s \mid \mathcal{E}_{\text{fact}})$, that could have given rise to the observed event. Then, for each plausible scene $s$, it calculates the expected outcome $\mu_i$ under that scene if the action were switched to $a'$. The final counterfactual prediction is the weighted average over all such plausible scenes.

Equation equation 3 suggests a direct *plug-in* estimation strategy: one could learn models for the outcome regression $\hat{\mu}_i$ and the scene posterior $\hat{p}(s \mid \mathcal{E}_{\text{fact}})$ and then approximate the integral using Monte Carlo methods. While this approach is intuitive, its reliance on a single, potentially misspecified outcome model is a significant limitation. In line with classical results, plug-in g-formula estimators can suffer from bias when the outcome regression is misspecified and may be statistically inefficient even when consistent Robins (1986); Tsiatis (2006).

To address these shortcomings, we use concepts from semiparametric efficiency theory Kennedy (2024); Schuler & van der Laan (2024), which provides methods for constructing estimators that are not only consistent but also have the smallest possible asymptotic variance, even when nuisance functions are estimated nonparametrically. A key technique from this work is the one-step estimator, which starts with an initial plug-in estimate and adds a correction term to reduce bias. This correction term is systematically derived from the estimand so-called efficient influence function (EIF) or canonical gradient under the nonparametric model. While these statistical tools are well-established, their application to retrospective inference from high-dimensional video data is not straightforward. Our work constructs a one-step estimator for our specific video-conditioned target quantity.

To build this estimator, we require three nuisance functions: the outcome regression model $\mu_i(a \mid a^{-i}, s)$, the action selection (propensity) model $\pi_a(s, a^{-i}) = P\big(A^i = a \mid A^{-i} = a^{-i}, S = s\big)$, and the conditional outcome probability model $\eta_{y \mid a}(s, a^{-i}) = P\big(Y = y \mid A^i = a, A^{-i} = a^{-i}, S = s\big)$. Using Bayes' rule, we define a *transport weight* that maps the distribution of episodes under the counterfactual action to our retrospective condition:

$$f(S; a', a_0, a^{-i}, y_0, v) = \frac{\eta_{y_0 \mid a_0}(S, a^{-i})\, \pi_{a_0}(S, a^{-i})\, p(v \mid S)}{\pi_{a'}(S, a^{-i})}. \tag{4}$$

Here, $p(v \mid S)$ is the conditional likelihood of observing video $v$ given scene state $S$. The resulting bias-corrected one-step estimator, $\hat{\psi}_i^{(bc)}$, is then:

$$\hat{\psi}_i^{(bc)}\big(a' \mid y_0, a_0, a^{-i}, v\big) = \frac{1}{n} \sum_{j=1}^{n} \frac{1}{\hat{e}_{\text{fact}}} \Big[ \underbrace{\mathbb{I}\{\mathcal{E}_{\text{fact},j}\}\, \hat{\mu}_i\big(a' \mid a^{-i}, S_j\big)}_{\text{Initial Plug-in Estimate}} \tag{5}$$

$$+ \underbrace{\mathbb{I}\big(A_j^i = a',\ A_j^{-i} = a^{-i}\big)\, \hat{f}\big(S_j; a', a_0, a^{-i}, y_0, v\big)\, \big(Y_j - \hat{\mu}_i\big(a' \mid a^{-i}, S_j\big)\big)}_{\text{Bias Correction Term}} \Big],$$

where the transport weight $\hat{f}$ is a plug-in estimate constructed from learned nuisance models. The second term in the summation is an empirical estimate of the EIF's contribution, designed to correct the initial estimate. This structure ensures a crucial property: the estimator remains consistent so long as either the outcome model (composed of $\hat{\mu}_i$ and $\hat{\eta}_{y|a}$) or the action selection model ($\hat{\pi}_a$) is correctly specified. Our one-step construction follows the standard influence-function blueprint: start from a plug-in target and add the empirical efficient influence function to remove first-order error, thereby attaining the semiparametric efficiency bound under regularity and consistent nuisance learning Tsiatis (2006); Bickel et al. (1993); van der Laan & Rose (2011). The exact EIF, a detailed derivation of this estimator, and a formal proof of its asymptotic properties are provided in the Appendix.

### 4.2 Learning Nuisance Components from Video

In this subsection, we describe how to learn the various nuisance functions required by the estimator in Equation equation 5. The entire process begins with learning an effective representation of the scene state $S$ from the pre-action video frames $V_0$.

**Learning the scene representation.** We decompose the scene state $S$ into two parts. The first is a set of discrete, semantically meaningful attributes $S^{\mathrm{d}}$ (e.g., object presence, categories, spatial relations). We extract these using a large multimodal model queried with structured prompts about the contents of $V_0$ Comanici et al. (2025), with outputs cross-checked by specialized perception models for tasks like segmentation and detection Kirillov et al. (2023b); Ravi et al. (2024); Liu et al. (2024b). The second is a continuous latent variable $S^{\mathrm{c}}$ that captures more nuanced aspects of the scene (e.g., lighting, continuous variations in object configurations). We learn a representation for $S^{\mathrm{c}}$ by training a Variational Autoencoder (VAE) Kingma & Welling (2013) on $V_0$, which maximizes the evidence lower bound (ELBO):

$$\mathcal{L}_{\mathrm{ELBO}}(\theta, \phi) = \mathbb{E}_{q_\phi(S^{\mathrm{c}}|V_0, S^{\mathrm{d}})}\big[\log P_\theta(V_0 \mid S^{\mathrm{c}}, S^{\mathrm{d}})\big] - \mathrm{KL}\big(q_\phi(S^{\mathrm{c}} \mid V_0, S^{\mathrm{d}}) \,\big\|\, P_\theta(S^{\mathrm{c}})\big).$$

The complete scene representation is the combination $S = (S^{\mathrm{d}}, S^{\mathrm{c}})$. During estimation, we can either sample from the learned posterior $q_\phi(\cdot \mid V_0, S^{\mathrm{d}})$ or use its mean as the scene representation for an event.

**Learning the nuisance functions.** Given the scene representation $S$, we can then learn the required nuisance functions: the outcome models $\hat{\mu}_i$ and $\hat{\eta}_{y|a}$, and the action selection model $\hat{\pi}_a$. These models can be trained using standard supervised learning techniques. For instance, when the outcome $Y$ is continuous, $\hat{\mu}_i$ can be trained using a regression model. When $Y$ is binary, it can be trained as a probabilistic classifier (e.g., via logistic regression). Since the action $A^i$ is discrete, both $\hat{\pi}_a$ and $\hat{\eta}_{y|a}$ can be framed as multi-class classification problems and trained with a loss function such as cross-entropy. A sufficient condition for our estimator to achieve its desirable statistical properties (e.g., semiparametric efficiency) is that these nuisance estimators converge at a rate of $o_P(n^{-1/4})$ Wasserman (2006); Chernozhukov et al. (2018). Many flexible non-parametric methods, such as kernel regression, random forests, or neural networks, can satisfy this condition under mild regularity assumptions. In our implementation, we employ cross-fitting to train these models, a procedure that helps prevent overfitting and satisfies a technical requirement for the estimator's theoretical guarantees.

## 5 Experiments

We conduct experiments using a controlled simulation to evaluate our proposed estimator. This allows for a precise quantitative analysis against the known data generating process, which is not possible with real-world video. The experiments are structured to first establish performance against baselines, then to probe robustness to model misspecification, and finally to analyze sensitivity to different causal structures.

### 5.1 Experimental Setup

**Data Generation.** We use a simulation based on a Structural Causal Model (SCM) to generate episodes. Each episode consists of a visual variable $V \in \mathbb{R}^{256 \times 256}$, a 3-dimensional scene state $S$,

co-actions $A^{-i} \in \{0,1\}^2$, a binary target action $A^i \in \{0,1\}$, and a binary outcome $Y \in \{0,1\}$. All variables are generated via logistic structural equations, as detailed in Appendix B. We generate 20 unique scenarios (8 non-confounding, 6 causal confounding, 6 non-causal confounding) that vary the strength and nature of the causal links. For example, in confounding scenarios (Figure 2 ii), the direct causal effect of $A^i$ on $Y$ is positive, while the confounding pathway through $(S, A^{-i})$ induces a negative correlation, making the task more challenging. For each scenario, we generate 50,000 samples with a 70/30 train/test split.

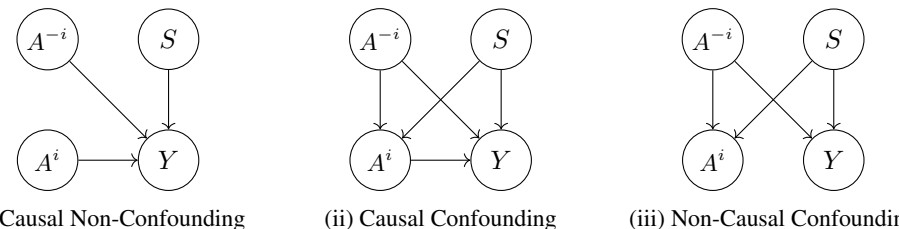

(i) Causal Non-Confounding      (ii) Causal Confounding      (iii) Non-Causal Confounding

Figure 2: The three causal structures used for evaluation. All potential causal links are shown; the absence of a link implies conditional independence.

**Baslines.** We compare our estimator against three baselines. **i) G-Formula**: A direct plug-in estimator based on Equation equation 3. **ii) Inverse Propensity Weighting (IPW)**: A standard method that corrects for confounding by re-weighting outcomes. The estimator for an average effect has the form $\mathbb{E}[Y(a)] = \mathbb{E}[\frac{\mathbb{I}(A=a)Y}{\pi_a(S)}]$. Our implementation adapts this concept for the bias correction term in our efficient estimator, as detailed in the Appendix. **iii) Fire-Zhu** Fire & Zhu (2016): This method searches for causal relationships by finding the action-outcome pair $(A, Y)$ that maximizes the information gain, or KL-divergence, $KL(P(A, Y) || P(A)P(Y))$. It essentially identifies the strongest statistical association.

**Evaluation Metric.** We evaluate all methods using the Mean Absolute Error (MAE) between the estimated counterfactual probability $\tilde{\psi}_i$ and the true counterfactual expectation which can be calculated by the SCM: $\text{MAE}(\tilde{\psi}_i) = \mathbb{E}\left[\left|\tilde{\psi}_i(a') - \mathbb{E}\left[Y^{(i)}(a') \mid Y = y, A^i = a_0, A^{-i}, V = v\right]\right|\right]$.

**Implementation Details.** All nuisance functions $(\hat{\mu}_i, \hat{\pi}_a, \hat{\eta}_{y|a})$ are implemented with logistic regression for the correctly specified setting. Model misspecification is induced by using incorrect functional forms, such as polynomial features or Random Forests. All models are trained using standard optimization libraries, and we employ cross-fitting to prevent overfitting and satisfy theoretical requirements for our estimator.

## 5.2 Performance Comparison with Baseline Methods

The purpose of this experiment is to evaluate the ability of each method to handle confounding when all nuisance models are correctly specified to match the true data generating process.

Table 1: Comparison of MAE across confounding scenarios with correctly specified models.

| Method | Non-Confounding | Confounding | Overall |
|---|---|---|---|
| Fire-Zhu | $0.0118 \pm 0.014$ | $0.6100 \pm 0.117$ | $0.3441 \pm 0.088$ |
| G-Formula | $0.0152 \pm 0.013$ | $0.0920 \pm 0.050$ | $0.0579 \pm 0.039$ |
| IPW | $0.0181 \pm 0.010$ | $0.1098 \pm 0.116$ | $0.0690 \pm 0.087$ |
| Our Estimator | $0.0185 \pm 0.011$ | $0.1155 \pm 0.132$ | $0.0724 \pm 0.100$ |

**Analysis of Results.** In the non-confounding case, the MAE of all methods is low, and the differences between them are smaller than their standard deviations, suggesting comparable performance. The introduction of confounding, however, reveals clear distinctions. The Fire-Zhu method,

which primarily measures correlation, fails when confounding is present, as it mistakes the spurious correlation for a causal link. In contrast, the other three methods effectively control for confounding because they explicitly use the scene state $S$ to adjust for it. Since all nuisance models are well-specified logistic functions matching the data generating process, they converge quickly. As a result, the performances of G-Formula, IPW, and our estimator are statistically similar given their variances.

## 5.3 Analysis of Robustness to Model Misspecification

This experiment investigates how estimators perform when their internal models are wrong, which is a common situation in practice. We induce misspecification by using incorrect functional forms for the outcome model ($\hat{\mu}_i$) or the propensity model ($\hat{\pi}_a$).

Table 2: Performance (MAE) when nuisance models are misspecified.

| Method | MAE w/ Misspec. | MAE w/o Misspec. | Degradation |
|---|---|---|---|
| G-Formula (Misspec) | $0.4163 \pm 0.041$ | $0.0693 \pm 0.065$ | +501.2% |
| IPW (Misspec) | $0.1877 \pm 0.233$ | $0.0656 \pm 0.056$ | +186.2% |
| Our Estimator w/ G-Formula Misspec | $0.0612 \pm 0.049$ | $0.0689 \pm 0.061$ | -11.2% |
| Our Estimator w/ IPW Misspec | $0.0674 \pm 0.063$ | $0.0689 \pm 0.061$ | -2.1% |

**Analysis of Results.** As shown in Table 2, misspecification causes the performance of both G-Formula and IPW to degrade considerably, consistent with their reliance on a single correctly specified model. In contrast, our estimator remains stable. This result empirically demonstrates the value of its structure. The bias correction term in Equation equation 5 allows the estimator to offset errors from one misspecified model by using information from the other, correctly specified model, providing a safeguard against inevitable modeling errors. The sensitivity of weighting-only strategies under model error and limited overlap is well-documented Kang & Schafer (2007).

## 5.4 Sensitivity to Confounding Structure

The previous experiment showed that IPW is less sensitive to misspecification than G-Formula. Here, we analyze if this is due to the underlying confounding structure. The pure IPW estimator for our retrospective target can be expressed as a weighted average:

$$\hat{\psi}_i^{(IPW)}(a') = \frac{\sum_j \mathbb{I}\{A_j^i = a', A_j^{-i} = a^{-i}\} \cdot \hat{f}_j \cdot Y_j}{\sum_j \mathbb{I}\{A_j^i = a', A_j^{-i} = a^{-i}\} \cdot \hat{f}_j}$$

This estimator's accuracy depends heavily on the transport weight $\hat{f}$. We can see from Equation equation 4 that $f$ depends on nuisance models involving the scene $S$. If the outcome $Y$ is conditionally independent of $S$ given the actions, $Y \perp\!\!\!\perp S \mid (A^i, A^{-i})$, then the terms involving $S$ in the numerator and denominator of $f$ may cancel, making the estimator less sensitive to how $S$ is modeled. This occurs in "action-only" confounding.

To test this, we define two confounding mechanisms. **i) Action-Only Confounding**: Confounders affect the action $A^i$ and outcome $Y$ through independent pathways. **ii) Joint Confounding**: Unobserved exogenous variables create complex dependencies between confounders, actions, and the outcome, violating the conditional independence assumption above.

Table 3: Performance (MAE) under different confounding structures and misspecification.

| Method | Action w/o Misspec | Action w/ Misspec | Degrade | Joint w/o Misspec | Joint w/ Misspec | Degrade |
|---|---|---|---|---|---|---|
| IPW | $0.0310 \pm 0.010$ | $0.0327 \pm 0.013$ | +5.3% | $0.1227 \pm 0.039$ | $0.5019 \pm 0.082$ | +308.9% |
| Our Estimator | $0.0302 \pm 0.009$ | $0.0328 \pm 0.014$ | +8.9% | $0.1517 \pm 0.036$ | $0.1379 \pm 0.030$ | -9.1% |

**Analysis of Results.** Under simple action-only confounding, both IPW and our estimator are robust to misspecification. However, joint confounding reveals a weakness in the IPW approach, where its error increases substantially. This supports our analysis: when confounding is complex, the transport weight $f$ becomes difficult to estimate accurately, and a misspecified propensity model leads to incorrect weights. By contrast, the influence-function correction targets the estimand's first-order error rather than relying solely on transport weights, which explains the observed stability under joint confounding Tsiatis (2006); van der Laan & Rose (2011).

## 6 CONCLUSION

In this work, we addressed the problem of inferring event-level counterfactuals from observational video. We formulated a retrospective causal question crucial for embodied agents, and developed a methodology that uses the observability of pre-action scenes to control for confounding. We proposed a statistically efficient estimator that integrates outcome and propensity models to provide robustness. Our simulation experiments showed that while standard methods are effective when models are well-specified, our estimator provides a necessary safeguard against the model misspecification that is common in complex, real-world settings.

## 7 DISCUSSION

The approach presented here has certain limitations, which also point to avenues for future work. A primary assumption is that all significant confounders are captured within the video frames. If a critical variable is unobserved, our method cannot account for its influence. This is less a theoretical flaw than a practical constraint on data collection; for physical systems, most confounders are, in principle, observable with sufficiently broad sensory input. Future work could explore methods for detecting the potential presence of unobserved confounders.

Additionally, the retrospective, event-conditioned nature of our estimator may require substantial data to find enough similar events for a stable estimate. While this is a valid concern, the ever-decreasing cost of video data collection suggests that for many applications, particularly for agents deployed continuously in an environment, this data requirement may be readily met.

## ETHICS STATEMENT

Our work adheres to the ICLR Code of Ethics. The research is based entirely on synthetic data, involves no human subjects, and we foresee no direct negative societal impacts or ethical concerns.

## REPRODUCIBILITY STATEMENT

To ensure reproducibility, we commit to making our source code and experimental configurations publicly available upon the acceptance of this paper.

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

# A    TECHNICAL APPENDIX

In this appendix, we introduce two results. First, we establish identification of the target under five assumptions (including theorem 3.4). Second, we derive the efficient influence function and prove that the one-step estimator $\hat{\psi}_i^{(bc)}$ is regular, asymptotically linear, and semiparametrically efficient in the nonparametric model.

## A.1    IDENTIFICATION

**Proposition A.1** (Identification under five assumptions). *Suppose theorems 3.1 to 3.5 hold. Let*

$$\mathcal{E}_{\text{fact}} := \{Y = y_0, A^i = a_0, A^{-i} = a^{-i}, V = v\}, \qquad \omega(s) := p(s \mid \mathcal{E}_{\text{fact}}), \qquad m(s) := \mathbb{E}\big[Y \mid A^i = a', A^{-i} = a^{-i}, S = s\big]$$

*Then*

$$\psi_i(a' \mid y_0, a_0, a^{-i}, v) = \int m(s)\,\omega(s)\,ds.$$

*Proof.* By the law of total expectation over $S$,

$$\psi_i = \int \mathbb{E}\Big[Y^{(i)}(a') \mid \mathcal{E}_{\text{fact}}, S = s\Big]\,\omega(s)\,ds.$$

Since $V \perp\!\!\!\perp (Y, A^i, A^{-i}) \mid S$ (theorem 3.5), $V$ is redundant once conditioning on $S$, hence the inner conditional reduces to $\mathbb{E}[Y^{(i)}(a') \mid Y = y_0, A^i = a_0, A^{-i} = a^{-i}, S = s]$. By consistency (theorem 3.1), $Y = Y^{(i)}(a_0)$ on $\{A^i = a_0\}$. By theorem 3.4, $Y^{(i)}(a') \perp\!\!\!\perp Y^{(i)}(a_0) \mid (S, A^{-i})$, thus

$$\mathbb{E}[Y^{(i)}(a') \mid Y = y_0, A^i = a_0, A^{-i} = a^{-i}, S = s] = \mathbb{E}[Y^{(i)}(a') \mid A^i = a_0, A^{-i} = a^{-i}, S = s].$$

By ignorability (theorem 3.3), $Y^{(i)}(a') \perp\!\!\!\perp A^i \mid (S, A^{-i})$, so the right-hand side equals $\mathbb{E}[Y^{(i)}(a') \mid A^{-i} = a^{-i}, S = s]$. A final application of consistency gives $\mathbb{E}[Y^{(i)}(a') \mid A^{-i} = a^{-i}, S = s] = \mathbb{E}[Y \mid A^i = a', A^{-i} = a^{-i}, S = s] = m(s)$, proving the claim. □

## A.2 Estimator efficiency

In this subsection we derive the efficient influence function and prove asymptotic linearity and efficiency of the one-step estimator. We introduce the transport weight

$$f(s) := \frac{\eta_{y_0|a_0}(s, a^{-i})\,\pi_{a_0}(s, a^{-i})\,p(v \mid s)}{\pi_{a'}(s, a^{-i})}, \qquad e_{\text{fact}} := P(\mathcal{E}_{\text{fact}}), \qquad \gamma(s) := \frac{f(s)}{e_{\text{fact}}}.$$

**Proposition A.2** (Efficient influence function). *Under the conditions of Proposition A.1, the efficient influence function (EIF) of $\psi_i$ is*

$$\phi(Z; \eta) = \frac{1}{e_{\text{fact}}}\Big[\mathbb{I}\{A^i = a', A^{-i} = a^{-i}\}\,f(S)\{Y - \mu_i(a' \mid a^{-i}, S)\} + \mathbb{I}\{\mathcal{E}_{\text{fact}}\}\,\mu_i(a' \mid a^{-i}, S)\Big] - \psi_i,$$

*and satisfies $P\phi = 0$.*

*Proof.* Write $\psi_i = \int m(s)\,\omega(s)\,ds$ with $m(s) = \mathbb{E}[Y \mid A^i = a', A^{-i} = a^{-i}, S = s]$ and $\omega(s) = p(s \mid \mathcal{E}_{\text{fact}})$. The influence-function map for products is linear. We compute the two components.

First, the IF of a pointwise conditional mean is (conditional-mean trick)

$$\text{IF}[m(s)] = \frac{\mathbb{I}\{A^i = a', A^{-i} = a^{-i}, S = s\}}{p(A^i = a', A^{-i} = a^{-i}, S = s)}\{Y - m(s)\}.$$

Therefore,

$$\int \text{IF}[m(s)]\,\omega(s)\,ds = \mathbb{I}\{A^i = a', A^{-i} = a^{-i}\}\,\frac{\omega(S)}{p(A^i = a', A^{-i} = a^{-i}, S)}\{Y - m(S)\}.$$

By Bayes and $V \perp\!\!\!\perp (Y, A^i, A^{-i}) \mid S$,

$$\frac{\omega(s)}{p(A^i = a', A^{-i} = a^{-i}, S = s)} = \frac{1}{e_{\text{fact}}} \cdot \frac{\eta_{y_0|a_0}(s, a^{-i})\,\pi_{a_0}(s, a^{-i})\,p(v \mid s)}{\pi_{a'}(s, a^{-i})} = \frac{f(s)}{e_{\text{fact}}},$$

hence the first integral equals $e_{\text{fact}}^{-1}\,\mathbb{I}\{A^i = a', A^{-i} = a^{-i}\}\,f(S)\{Y - m(S)\}$.

Second, the IF of the conditional law $\omega(s) = \mathbb{E}[\mathbb{I}\{S = s\} \mid \mathcal{E}_{\text{fact}}]$ is

$$\text{IF}[\omega(s)] = \frac{\mathbb{I}\{\mathcal{E}_{\text{fact}}\}}{e_{\text{fact}}}\left(\mathbb{I}\{S = s\} - \omega(s)\right).$$

Therefore,

$$\int m(s)\,\text{IF}[\omega(s)]\,ds = \frac{\mathbb{I}\{\mathcal{E}_{\text{fact}}\}}{e_{\text{fact}}}\,m(S) - \psi_i.$$

Summing the two integrals yields the stated $\phi(Z; \eta)$. The mean-zero property $P\phi = 0$ follows by conditioning on $(S, A^{-i}, A^i)$ and using $\mathbb{E}[Y - m(S) \mid S, A^{-i}, A^i = a'] = 0$ together with $\mathbb{E}[m(S) \mid \mathcal{E}_{\text{fact}}] = \psi_i$. $\square$

**Proposition A.3** (Asymptotic linearity and semiparametric efficiency). *Let $(\hat{\mu}_i, \hat{\eta}_{y|a}, \hat{\pi}_a, \widehat{p}(v \mid s), \hat{e}_{\text{fact}})$ be cross-fitted nuisance estimates, and define $\hat{f}$ and $\hat{\gamma} := \hat{f}/\hat{e}_{\text{fact}}$. The one-step estimator $\hat{\psi}_i^{(bc)}$ in eq. (5) satisfies*

$$\sqrt{n}\big(\hat{\psi}_i^{(bc)} - \psi_i - \mathbb{P}_n\phi\big) \xrightarrow{P} 0, \qquad \sqrt{n}\big(\hat{\psi}_i^{(bc)} - \psi_i\big) \rightsquigarrow \mathcal{N}\big(0, P[\phi^2]\big),$$

*provided*

$$\|\hat{\gamma} - \gamma\|_{L_2(P_{S|A^i=a', A^{-i}=a^{-i}})} = o_P(n^{-1/4}),$$

$$\|\hat{\mu}_i - \mu_i\|_{L_2(P_{S|A^i=a', A^{-i}=a^{-i}})} = o_P(n^{-1/4}).$$

*Consequently, $\hat{\psi}_i^{(bc)}$ is semiparametrically efficient in the stated nonparametric model.*

*Proof.* Let $\hat{\phi}$ be Proposition A.2 with nuisances replaced by estimates, and let $\hat{\psi}_i$ denote the direct plug-in for equation 3. Decompose

$$\hat{\psi}_i^{(bc)} - \psi_i - \mathbb{P}_n\phi = \underbrace{(\hat{\psi}_i - \psi_i + P\hat{\phi})}_{\text{(I)}} + \underbrace{(\mathbb{P}_n - P)(\hat{\phi} - \phi)}_{\text{(II)}}.$$

By cross-fitting and $\|\hat{\phi} - \phi\|_{L_2(P)} \to 0$, term (II) is $o_P(n^{-1/2})$. It remains to control (I). A direct calculation using Bayes' identity $\omega(s) = \gamma(s)\,p(A^i = a', A^{-i} = a^{-i}, S = s)$ yields

$$\text{(I)} = P(A^i = a', A^{-i} = a^{-i})\left\langle \hat{\gamma} - \gamma, \; \mu_i - \hat{\mu}_i \right\rangle_{L_2(P_{S|A^i=a',A^{-i}=a^{-i}})} + \left(\frac{e_{\text{fact}}}{\hat{e}_{\text{fact}}} - 1\right)\mathbb{E}_{S\sim\omega}\left[\hat{\mu}_i(a' \mid a^{-i}, S)\right].$$

By Cauchy–Schwarz and the assumed rates, the inner product is $o_P(n^{-1/2})$; the prefactor remainder is $O_P(n^{-1/2})$ when $\hat{e}_{\text{fact}}$ is the empirical frequency of $\mathcal{E}_{\text{fact}}$. Therefore (I) $= o_P(n^{-1/2})$, which implies the claimed asymptotic linearity. Efficiency follows because the one-step estimator is regular with influence function equal to the EIF. $\qquad\square$

### A.3 DERIVATION OF AN IPW-STYLE ESTIMATOR

Here, we provide two derivations for an IPW-style estimator for our target quantity. This type of estimator relies solely on a re-weighting scheme and does not use an outcome model.

**Derivation 1: From the One-Step Estimator.** A straightforward way to derive a pure re-weighting estimator is to set the outcome model $\hat{\mu}_i$ to zero in our main estimator, Equation equation 5. In this case, the first term disappears (as $\hat{\mu}_i = 0$ inside the sum over the retrospective event, which has measure zero for continuous variables). The estimator becomes:

$$\hat{\psi}_i^{(IPW,1)}\left(a' \mid y_0, a_0, a^{-i}, v\right) = \frac{1}{n}\sum_{j=1}^{n}\frac{1}{\hat{e}_{\text{fact}}}\mathbb{I}\left(A_j^i = a', A_j^{-i} = a^{-i}\right)\hat{f}\left(S_j; a', a_0, a^{-i}, y_0, v\right)Y_j$$

$$\tag{6}$$

This can be rewritten as the familiar weighted average form presented in Section 5.4.

**Derivation 2: Direct Reweighting.** Alternatively, we can derive the weights directly. Following the logic in the provided draft ('formulation.tex') and adapting to our notation, we seek a weight function $w_j = \mathbb{I}\{A_j^i = a', A_j^{-i} = a^{-i}\} \cdot f(S_j)$ such that the weighted average of the outcome $Y$ equals our target quantity. This requires finding a function $f(S)$ that satisfies:

$$\mathbb{E}[wY] = \psi_i(a')$$

Expanding the left side and matching terms with the identified expression for $\psi_i$ (Proposition A.1) leads to the same transport weight $f(S)$ as defined in Equation equation 4. This confirms that the transport weight is the correct function to map the distribution of outcomes under the counterfactual action to the desired retrospective conditional expectation.

### A.4 DERIVATION OF THE IPW-STYLE ESTIMATOR

We derive an Inverse Probability Weighting (IPW) estimator for the target quantity $\psi_i(a' \mid y_0, a_0, a^{-i}, v)$. The goal is to find an observable weight function, $w$, such that the weighted expectation $\mathbb{E}[wY]$ equals the identified estimand $\psi_i$. From Proposition A.1, the identified estimand is:

$$\psi_i = \int m(s)\,\omega(s)\mathrm{d}s,$$

where $m(s) := \mathbb{E}[Y \mid A^i = a', A^{-i} = a^{-i}, S = s]$ and $\omega(s) := p(s \mid \mathcal{E}_{\text{fact}})$. We propose a weight of the form $w = \mathbb{I}\{A^i = a', A^{-i} = a^{-i}\} \cdot \gamma(S)$, where $\gamma(S)$ is a function to be determined.

**Derivation of the Weight Function**  We set the expectation of the weighted outcome equal to the target estimand, $\mathbb{E}[wY] = \psi_i$, and solve for $\gamma(S)$.

The left-hand side (LHS) is expanded using the law of total expectation:

$$\begin{aligned}
\mathbb{E}[wY] &= \mathbb{E}\left[\mathbb{I}\{A^i = a', A^{-i} = a^{-i}\}\gamma(S)Y\right] \\
&= \mathbb{E}\left[\gamma(S)\mathbb{I}\{A^i = a', A^{-i} = a^{-i}\}\mathbb{E}[Y \mid S, A^i, A^{-i}]\right] \\
&= \mathbb{E}\left[\gamma(S)\mathbb{I}\{A^i = a', A^{-i} = a^{-i}\}m(S)\right] \\
&= \int m(s)\gamma(s)p(A^i = a', A^{-i} = a^{-i}, S = s)\mathrm{d}s
\end{aligned}$$

The right-hand side (RHS) is $\psi_i = \int m(s)p(s \mid \mathcal{E}_{\text{fact}})\mathrm{d}s$.

Equating the integrands (for any arbitrary outcome model $m(s)$) yields the condition:

$$\gamma(s)p(A^i = a', A^{-i} = a^{-i}, S = s) = p(s \mid \mathcal{E}_{\text{fact}})$$

Solving for $\gamma(s)$ gives:

$$\gamma(s) = \frac{p(s \mid \mathcal{E}_{\text{fact}})}{p(A^i = a', A^{-i} = a^{-i}, S = s)}$$

Using Bayes' rule and the definitions $\pi_{a'}(s, a^{-i}) := p(A^i = a' \mid A^{-i} = a^{-i}, S = s)$ and $e_{\text{fact}} := P(\mathcal{E}_{\text{fact}})$, we expand the numerator and denominator:

$$\begin{aligned}
\gamma(s) &= \frac{p(\mathcal{E}_{\text{fact}} \mid S = s)p(s)/p(\mathcal{E}_{\text{fact}})}{p(A^i = a' \mid A^{-i} = a^{-i}, S = s)p(A^{-i} = a^{-i}, S = s)} \\
&= \frac{p(\mathcal{E}_{\text{fact}} \mid S = s)p(s)}{e_{\text{fact}} \cdot \pi_{a'}(s, a^{-i})p(A^{-i} = a^{-i} \mid s)p(s)}
\end{aligned}$$

The term $p(\mathcal{E}_{\text{fact}} \mid S = s)$ expands to $\eta_{y_0 \mid a_0}(s, a^{-i})\,\pi_{a_0}(s, a^{-i})\,p(A^{-i} = a^{-i} \mid s)\,p(v \mid s)$ under the stated assumptions. After cancelling the term $p(A^{-i} = a^{-i} \mid s)p(s)$, we obtain:

$$\gamma(s) = \frac{1}{e_{\text{fact}}} \cdot \frac{\eta_{y_0 \mid a_0}(s, a^{-i})\,\pi_{a_0}(s, a^{-i})\,p(v \mid s)}{\pi_{a'}(s, a^{-i})}$$

This confirms that $\gamma(s) = f(s)/e_{\text{fact}}$, matching the definition of the transport weight $f(s)$ in Appendix A.2.

The IPW estimator for $\psi_i$ is the empirical analogue of $\mathbb{E}[wY]$:

$$\hat{\psi}_i^{(IPW)} = \frac{1}{n}\sum_{j=1}^{n} w_j Y_j = \frac{1}{n}\sum_{j=1}^{n}\mathbb{I}\{A_j^i = a', A_j^{-i} = a^{-i}\}\hat{\gamma}(S_j)Y_j$$

Substituting the derived expression for $\gamma(S)$ yields the final form:

$$\hat{\psi}_i^{(IPW)}\left(a' \mid y_0, a_0, a^{-i}, v\right) = \frac{1}{n\hat{e}_{\text{fact}}}\sum_{j=1}^{n}\mathbb{I}\left(A_j^i = a', A_j^{-i} = a^{-i}\right)\hat{f}(S_j)\,Y_j$$

where $\hat{f}$ and $\hat{e}_{\text{fact}}$ are estimates of the transport weight and the probability of the factual event, respectively.

## B  STRUCTURAL EQUATIONS FOR EXPERIMENTAL SCMS

This appendix provides the complete structural equations for the five Structural Causal Models (SCMs) used in our experimental evaluation. These models share a common generative process rooted in visual information.

First, the pre-action video frames $V$ are processed by a pretrained encoder to produce a feature vector. This vector is then partitioned into three disjoint segments: $(S_{A^i}, S_{A^{-i}}, S_Y)$, which represent the exogenous scene variables that can influence the target action, co-actions, and the outcome.

Different confounding structures are created by specifying which of these scene components act as causes for each variable.

All stochasticity originates from independent exogenous noise variables $U_{A^i}, U_{A^{-i}}, U_Y$, each drawn from a standard logistic distribution, $U \sim \text{Logistic}(0, 1)$. For clarity, the structural equations below are presented in a general functional form, such as $A^i = f_{A^i}(\text{causes}, U_{A^i})$. In our implementation, these functions $f(\cdot)$ are realized using a linear model of the causes, with the final binary outcome determined by whether this linear combination exceeds the threshold set by the logistic noise variable. This is equivalent to sampling from a Bernoulli distribution whose probability is the sigmoid of the linear combination.

## B.1 Model 1: Causal Non-Confounding

This baseline model features a direct causal effect from $A^i$ to $Y$. There is no confounding from the scene, as each scene component only influences its corresponding variable.

$$(S_{A^i}, S_{A^{-i}}, S_Y) = \text{partition}(\text{encode}(V)) \tag{7}$$
$$U_{A^i}, U_{A^{-i}}, U_Y \sim \text{Logistic}(0, 1) \tag{8}$$
$$A^{-i} = f_{A^{-i}}(S_{A^{-i}}, U_{A^{-i}}) \tag{9}$$
$$A^i = f_{A^i}(U_{A^i}) \tag{10}$$
$$Y = f_Y(A^i, U_Y) \tag{11}$$

Key characteristics: $A^i$ is independent of any scene component, and $Y$ is independent of any scene component given $A^i$.

## B.2 Model 2: Causal Confounding

Here, the scene confounds the relationship between the target action $A^i$ and outcome $Y$. The scene component for $Y$, $S_Y$, influences $A^i$, and vice-versa. The direct causal path $A^i \to Y$ is maintained.

$$(S_{A^i}, S_{A^{-i}}, S_Y) = \text{partition}(\text{encode}(V)) \tag{12}$$
$$U_{A^i}, U_{A^{-i}}, U_Y \sim \text{Logistic}(0, 1) \tag{13}$$
$$A^{-i} = f_{A^{-i}}(S_{A^{-i}}, U_{A^{-i}}) \tag{14}$$
$$A^i = f_{A^i}(S_{A^i}, S_Y, U_{A^i}) \tag{15}$$
$$Y = f_Y(A^i, S_{A^i}, S_Y, U_Y) \tag{16}$$

Confounding structure: The arguments of the functions show that scene variables create paths such as $A^i \leftarrow S_Y \to Y$ and $A^i \leftarrow S_{A^i} \to Y$.

## B.3 Model 3: Causal Confounding (Joint)

This model introduces a denser confounding structure where all variables are influenced by all scene components, creating complex dependencies while preserving the direct causal effect of $A^i$ on $Y$.

$$(S_{A^i}, S_{A^{-i}}, S_Y) = \text{partition}(\text{encode}(V)) \tag{17}$$
$$U_{A^i}, U_{A^{-i}}, U_Y \sim \text{Logistic}(0, 1) \tag{18}$$
$$A^{-i} = f_{A^{-i}}(S_{A^i}, S_{A^{-i}}, S_Y, U_{A^{-i}}) \tag{19}$$
$$A^i = f_{A^i}(S_{A^i}, S_{A^{-i}}, S_Y, U_{A^i}) \tag{20}$$
$$Y = f_Y(A^i, S_{A^i}, S_{A^{-i}}, S_Y, U_Y) \tag{21}$$

### B.4 MODEL 4: NON-CAUSAL CONFOUNDING

This model is designed to test for false positives. There is no direct causal effect from $A^i$ to $Y$. Any observed correlation is spurious, induced by the same confounding structure as in Model 2.

$$(S_{A^i}, S_{A^{-i}}, S_Y) = \text{partition}(\text{encode}(V)) \tag{22}$$
$$U_{A^i}, U_{A^{-i}}, U_Y \sim \text{Logistic}(0, 1) \tag{23}$$
$$A^{-i} = f_{A^{-i}}(S_{A^{-i}}, U_{A^{-i}}) \tag{24}$$
$$A^i = f_{A^i}(S_{A^i}, S_Y, U_{A^i}) \tag{25}$$
$$Y = f_Y(S_{A^i}, S_Y, U_Y) \tag{26}$$

Critical difference: The function $f_Y$ does not take $A^i$ as an argument, formalizing that $Y \perp A^i \mid (S_{A^i}, S_Y)$.

### B.5 MODEL 5: NON-CAUSAL CONFOUNDING (JOINT)

Similar to Model 4, there is no direct causal path from $A^i$ to $Y$. The spurious association is generated by the dense, joint confounding structure from Model 3.

$$(S_{A^i}, S_{A^{-i}}, S_Y) = \text{partition}(\text{encode}(V)) \tag{27}$$
$$U_{A^i}, U_{A^{-i}}, U_Y \sim \text{Logistic}(0, 1) \tag{28}$$
$$A^{-i} = f_{A^{-i}}(S_{A^i}, S_{A^{-i}}, S_Y, U_{A^{-i}}) \tag{29}$$
$$A^i = f_{A^i}(S_{A^i}, S_{A^{-i}}, S_Y, U_{A^i}) \tag{30}$$
$$Y = f_Y(S_{A^i}, S_{A^{-i}}, S_Y, U_Y) \tag{31}$$

As in Model 4, the absence of $A^i$ as an argument in $f_Y$ signifies no direct causation.

### B.6 PARAMETER SPECIFICATIONS

To ensure a robust evaluation, the coefficients $\{\beta, \alpha\}$ for the underlying linear models that implement the functions $f(\cdot)$ are sampled from distributions with significant variance.

**Baseline Parameters (Intercepts):**

- The intercept terms $\alpha$ for all functions are sampled from $\mathcal{N}(0, 0.5^2)$.

**Causal Effect Parameters:**

- The coefficient for $A^i$ in the linear model for $f_Y$, denoted $\beta_Y^{A^i}$, is sampled from $\mathcal{N}(3.0, 1.2^2)$ for causal scenarios (Models 1, 2, 3).
- $\beta_Y^{A^i} = 0$ for non-causal scenarios (Models 4, 5).

**Confounding Parameters (Scene-to-Variable Effects):**

- Coefficients for scene effects (e.g., the weight of $S_Y$ in the model for $f_{A^i}$) are sampled from distributions with large means and variances, such as $\mathcal{N}(6.0, 2.0^2)$, to create strong confounding.

**Interaction Strengths and Other Effects:**

- Coefficients for joint confounding (e.g., the weight of $S_{A^{-i}}$ in the model for $f_{A^i}$) use smaller, but non-trivial, variances (e.g., $\mathcal{N}(0.5, 0.4^2)$) to ensure complex but identifiable models.

This parameterization scheme tests the methods' performance across a wide range of data-generating conditions.

## C    USE OF LARGE LANGUAGE MODELS

We declare that a large language model was utilized solely for improving the grammar and clarity of this manuscript. All core scientific contributions are our own.

