# OpenReview forum: "Video Causal Understanding with Scene-conditioned Counterfactuals"
_ICLR.cc/2026/Conference — Submitted to ICLR 2026_

### Official Review · Reviewer_SnZU · 2025-10-23

**Soundness:** 3
**Presentation:** 3
**Contribution:** 3
**Rating:** 6
**Confidence:** 2

**Summary:**

The paper addresses the challenge of determining true causal relationships between actions and outcomes in video data, where confounding visual factors often lead to spurious correlations.
To tackle this, the authors reformulate video causal reasoning as a scene-conditioned counterfactual inference problem — asking, “In this specific scene, what would have happened if the action were different?” They propose a doubly robust, semiparametric estimator that integrates both outcome and propensity models to adjust for confounding learned from video frames. This estimator remains consistent even if one of the models is misspecified, providing robustness against real-world uncertainty.
Experiments in controlled simulation environments demonstrate that the proposed method yields more accurate and stable causal attributions compared to traditional approaches such as G-formula, IPW.

**Strengths:**

1. The research topic is interesting, and there remains a significant gap in the study of video causality.

2. The authors take into account the high-dimensional nature of video data and construct a one-step estimator for their specific video-conditioned target quantity.

**Weaknesses:**

Although the paper explains how confounding causal relationships in videos can be eliminated, it remains difficult for researchers—especially those in the computer vision field—to translate these insights or methods into their deep learning models (e.g., video large language models).

**Questions:**

Could the authors summarize in more accessible language how the one-step estimator is adapted to high-dimensional video data, which differs from traditional statistical data?

---

### Official Review · Reviewer_aq87 · 2025-11-01

**Soundness:** 2
**Presentation:** 3
**Contribution:** 2
**Rating:** 4
**Confidence:** 2

**Summary:**

This paper proposes a framework for scene-conditioned counterfactual reasoning in video, aiming to estimate what the outcome of an event would have been if a specific action were changed while keeping the rest of the scene fixed. The authors formalize the problem under a structural causal model, define the event-level counterfactual mean, and derive a one-step estimator that achieves double robustness and asymptotic efficiency. They also propose a hybrid scene representation combining discrete structured variables and continuous latent variables, and validate the method on synthetic environments generated from hand-crafted causal models.

**Strengths:**

- This paper presents a clear causal formulation of event-level counterfactual reasoning, grounded in potential outcomes and semiparametric efficiency theory.
- The one-step bias-corrected estimator is rigorous and well-motivated, offering statistical robustness to model misspecification.
- The paper is well-structured and theoretically consistent, with assumptions (e.g., ignorability, scene–video separability) clearly stated.
- It contributes theoretical groundwork for causal estimation in visual or sequential domains.

**Weaknesses:**

My main concern with this paper is their experiment. The whole experiments are based on fully synthetic data, without any "video" dataset. I am not familiar with this field and do not know if this is the standard. I checked several related works cited by the authors, and they all used real video data. My opinion is, if it does not validate on real-world or synthetic video data, it is not fair to claim the contribution with "video understanding". In fact, here V can be anything. However, I would like to see how other reviewers think and how the authors respond during the rebuttal.
- The work is not evaluated on real or visual video data. All experiments use synthetic samples generated from simulated structural causal models.
- The “video” variable is not an actual video but a numeric proxy derived from a latent 3D scene vector, with no pixel-level or temporal information.
- Assumption 5 (scene–video separability) is strong and untested, and the simulated scene state does not reflect real-world visual complexity.
- There is no analysis of how the proposed estimator would scale to high-dimensional or temporally extended inputs, which is central to real-world video understanding.
- The motivation of this paper is strong, as illustrated in Figure 1, and the problem it aims to address is important and timely. However, after reading the paper, I do not think the authors provide sufficient experimental evidence to support their claims. The experiments are entirely synthetic and do not involve real or visually grounded video data. As a result, it remains unclear whether the proposed method can actually improve counterfactual reasoning or causal understanding in realistic video scenarios. The current results demonstrate theoretical validity but not practical effectiveness.

**Questions:**

- The paper’s experiments are entirely synthetic and do not use real or visually grounded video data. How do the authors envision applying their method to real videos with pixel-level and temporal complexity?
- Can the authors clarify whether their framework could be integrated with pretrained video encoders to extract the latent scene state S instead of simulating it?
- Assumption 5 is quite strong. How realistic is this assumption in real-world video settings? Have the authors explored ways to relax or empirically test it?
- How sensitive is the proposed estimator to errors in scene representation? For instance, if S were learned imperfectly from a video encoder, how would this affect counterfactual estimation?
- Since the “video” variable V in the experiments is only a simulated vector, what evidence supports the claim that this framework advances video causal understanding?
- Recent works such as "LLCP: Learning Latent Causal Processes for Reasoning-Based Video Question Answering" directly use real videos and learn causal representations from visual input. In contrast, this paper relies entirely on synthetic, low-dimensional simulations. Could the authors clarify why they did not test their method on any real or visually grounded video datasets?

---

### Official Review · Reviewer_HweA · 2025-11-03

**Soundness:** 2
**Presentation:** 3
**Contribution:** 2
**Rating:** 4
**Confidence:** 4

**Summary:**

The paper provides a theoretical solution for causal understanding in videos, emphasizing event-level reasoning rather than population-averaged effects.

**Strengths:**

Experiments generate simulated data via Structural Causal Models (SCMs), enabling quantitative comparison with ground truth and addressing the limitation of real video data lacking definitive ground truth.

**Weaknesses:**

1. The core assumption of the paper is that all confounders are contained in video frames (i.e., the ignorability assumption holds). However, unobserved confounders may exist in practice (e.g., external environmental changes or internal states of the agent). The paper does not discuss how to detect or mitigate this limitation, which may affect the method’s reliability in open-world scenarios. It is recommended to add sensitivity analysis or robustness checks to evaluate the potential impact of unmeasured confounding.

2. The method involves multiple components (e.g., scene representation learning, nuisance model training, and cross-fitting), which may lead to high computational costs—especially for long videos or large-scale data. The paper does not provide runtime or resource requirement analysis, and VAE training may require extensive annotation or parameter tuning. In real-world deployment, this could limit its application to resource-constrained scenarios (e.g., edge devices).

3. Although the simulated experiments are rigorously designed, reliance on synthetic data may fail to fully capture the complexity of real videos (e.g., noise, occlusion, or varied lighting). The paper has not been validated on public real-video datasets (e.g., motion or driving videos), leaving its generalization ability questionable. Additionally, experiments primarily target binary actions and outcomes, without extending to continuous or multi-class settings—reducing the method’s versatility.

4. The paper relies on strong assumptions, such as positivity (i.e., all actions have non-zero probabilities under a given scene). However, limited overlap in real-world data may lead to unstable estimators. The paper does not discuss how to address such issues (e.g., trimming or regularization), which may hinder practical application. Meanwhile, the quality of scene representation learning is highly dependent on foundation models; if extraction fails (e.g., due to model bias), errors may be introduced.

**Questions:**

Please see the weaknesses.

---

### Official Review · Reviewer_8PbF · 2025-11-03

**Soundness:** 3
**Presentation:** 3
**Contribution:** 3
**Rating:** 6
**Confidence:** 3

**Summary:**

This paper formulates a novel task of scene-conditioned counterfactual inference from observational videos, aiming to estimate the causal effect of a specific action on an outcome. To address this problem, the paper proposes an efficient and doubly robust estimator that integrates outcome and propensity models. Experiments conducted on a synthetic dataset demonstrate the robustness in the confounding scenarios and model misspecification scenarios.

**Strengths:**

1. This paper formally defines the task of video causal understanding as a scene-conditioned counterfactual inquiry. It establishes a theoretically-grounded estimation framework for this problem, complete with clear identification assumptions and derivations of the estimator.

2. The paper proposes an efficient and doubly robust estimator. This design ensures consistent estimation of the causal effect, remaining reliable even if one of the underlying models (either the outcome or the propensity model) is misspecified.

3. Through systematic experiments under confounding scenarios and model misspecification scenarios, the paper exhibits the superior and robust performance of the proposed estimator.

**Weaknesses:**

1. The performance relies on the quality of the scene confounder representation $S$. Although a hybrid representation is proposed and cross-checking is mentioned, there is no ablation study or quantitative analysis to show its contribution.

2. The set of baseline methods used for comparison is relatively small. Furthermore, as shown in Table 1, the performance advantage of the proposed method over baselines like G-Formula and IPW under correct specification is not pronounced.

3. The experimental evaluation is conducted solely on synthetic data. The absence of validation on real-world video datasets weakens the claim about the method's ability to capture complex confounders in practical scenarios and limits its persuasiveness.

**Questions:**

1. There is a recurring typo in the manuscript where the word "equation" appears twice (e.g., "Equation equation 1") in several formula references, which should be corrected.

2. Could the authors comment on the computational cost of their framework, particularly the scene representation learning and the one-step estimation procedure?

---

### Official Review · Reviewer_qNNy · 2025-11-05

**Soundness:** 3
**Presentation:** 3
**Contribution:** 2
**Rating:** 4
**Confidence:** 3

**Summary:**

The paper formulates scene-conditioned and event-level counterfactuals for video causal understanding. Given a realized episode, it estimates the expected outcome if a target action had been different. It introduces a one-step, doubly robust estimator that combines an outcome model and a propensity model. On a simulation dataset, the proposed method achieved SOTA performance.

**Strengths:**

- Novelty: Clear formulation of scene-conditioned counterfactuals at the event level, distinct from ATE/CATE.
- Quality: Identification under stated assumptions and principled EIF-based one-step estimator with double robustness and efficiency proof.
- Strong performance: Empirical analysis isolates confounding structures and shows robustness to outcome/propensity misspecification compared with plug-in and IPW.

**Weaknesses:**

- Lack of comparison in real-world data: All results are on synthetic SCMs. Adding experiments on at least controlled real dataset or video QA/embodied can demonstrate its practicality of the proposed approach.
- Strong model assumption: Scene–video separability (V ⊥ (Y,A) | S) can be strong for real videos. It'll be good to analyze its sensitivity to violations.
- Lack of learning details: It's unclear how p(v|S) and the hybrid scene state (Sd, Sc) are implemented and validated.
- Lack of ablation studies on scene representation: It'll be good to include ablations over detectors/segmenters, latent dimension, and prompt variants.

**Questions:**

- Scene representation robustness: What is the sensitivity to Sd detection errors and Sc latent dimension?
- Computational cost: Report training and inference time for nuisance models and estimator on your largest setting; discuss scalability to longer videos and larger S.

---

### Meta-Review · Area_Chair_Rvck · 2026-01-01

**Summary:**

The authors did not provide a response during the rebuttal phase. As a result, the reviewers’ concerns, such as the lack of experiments on real-world data and the reliance on strong assumptions for identifying counterfactual events, remain unaddressed. Consequently, the paper is not ready for publication in its current form.

**Reviewer Concerns:**

As no rebuttal was provided, the reviewers’ concerns remain outstanding.

**Reviewer Scores:**

No score change since no rebuttal was provided.

---

### Decision · Program_Chairs · 2026-01-26

Reject